# Effects of Temperature and Host Plant on Hedgehog Grain Aphid, *Sipha maydis* Demographics

**DOI:** 10.3390/insects14110862

**Published:** 2023-11-08

**Authors:** Mason Taylor, Rafael Hayashida, William Wyatt Hoback, John Scott Armstrong

**Affiliations:** 1Department of Entomology and Plant Pathology, Oklahoma State University, Stillwater, OK 74078, USA; masonta@okstate.edu (M.T.); rafael.hayashida@okstate.edu (R.H.); 2U.S. Department of Agriculture, Agricultural Research Service, Wheat, Peanut and Other Field Crops Research Unit, 1301 North Western Road, Stillwater, OK 74075, USA; scootamous@gmail.com

**Keywords:** lifetable, fecundity, resistance, exotic aphid, generalist

## Abstract

**Simple Summary:**

The hedgehog grain aphid (HGA) is a pest that affects cereal crops worldwide and was first observed in the United States in 2007. To understand how this aphid spreads and becomes damaging, it is essential to study how temperature and the host plant affect its development and reproduction. In this study, we found that temperatures between 20 °C and 25 °C are best for HGA survival and reproduction, while temperatures below 10 °C and above 35 °C do not allow for survival. This study revealed that HGA can survive on wheat, millet, and three cultivars of sorghum, including those resistant to the sorghum aphid. Together, these results help us predict HGA populations by identifying host plants in cereal crops and documenting how temperature affects these populations.

**Abstract:**

The hedgehog grain aphid (HGA), *Sipha maydis* Passerini (Hemiptera: Aphididae), is a cereal pest in many regions of the world. It was first documented in the United States in 2007, and it has a range that appears to be expanding. Understanding the effects of temperature and the host plant on HGA development, survival, and reproduction is crucial for understanding its population dynamics, potential distribution, and management strategies. In this study, we investigated the effects of different temperatures and host plants on the demographic parameters of HGA and determined the supercooling point (SCP) for their first instars, apterous adults, and winged adults. Our findings revealed that temperatures between 20 °C and 25 °C were optimal for HGA development and reproduction, with parthenogenetic females producing approximately 60 offspring in their lifetimes. However, HGA development was hindered below 10 °C and above 35 °C. The SCP for HGA was similar (mean ± S.E.: −16.280 ± 0.532 °C) among nymphs, apterous adults, and winged adults. We compared the HGA demographics with the demographics of the sorghum aphid (SA), *Melanaphis sorghi* (Theobald, 1904), on wheat, millet, and three cultivars of sorghum under a constant temperature. The HGA completed its life cycle on all the tested host plants with a similar reproduction, demonstrating a lack of resistance to HGA by a sorghum that is resistant to SA. By expanding our knowledge of host plant- and temperature-dependent development, reproduction, and mortality in *S. maydis*, we can better predict and manage future HGA populations in small grain crops.

## 1. Introduction

The hedgehog grain aphid (HGA), *Sipha maydis* Passerini (Hemiptera: Aphididae), is a cereal pest in many regions of the world [1,2,3,4,5]. It feeds on a variety of wild grasses and cereal crops and has been documented to use 52 plant host species worldwide, including wheat (*Triticum aestivum*), sorghum (*Sorghum bicolor*), barley (*Hordeum vulgare*), and wild grasses [2,6]. HGA was first detected in the United States in 2007, where it was found feeding on giant wild rice, *Leymus condensatus* [7]. The most recent field assessment, conducted from 2015 to 2017, revealed that HGA has since expanded, occurring in New Mexico, Colorado, Utah, and Wyoming, where it uses a range of grass species [2]. Since this first observation, HGA has been monitored because of its capacity to cause damage to crops [1,3]. Damage occurs through both direct injury and by transmitting plant viruses [8,9,10].

Controlling aphid pests presents a significant challenge due to the distinctive features of aphid biology. This includes their cyclic parthenogenic reproduction, the ability to disperse actively or passively through flight, and their increasing resistance to insecticides [8]. Furthermore, this difficulty is compounded by their widespread distribution, wide range of host plants, and the adaptability to various climatic conditions exhibited by HGA. HGA remains a persistent threat throughout the growth stages of plants, from seedlings to full maturation, making effective pest management particularly demanding [1,3].

Temperature plays a crucial role in aphid population dynamics, as external environmental conditions influence all aspects of aphid biology [11,12,13]. The impact of temperature on aphid fitness can be observed through their reproduction, growth, survival, and rate of population increase [13,14,15]. There is often an optimal temperature range in which aphids have the highest rate of population increase, and this temperature varies depending on the species of aphid and the origin of the population [14]. Knowing the optimal temperature range, as well as the upper and lower developmental thresholds for survival and reproduction, are essential for predicting future outbreaks and determining the potential distribution limits for HGA. Additionally, documenting the supercooling point (SCP) can allow for prediction of HGA distribution [16,17], distinguishing areas where recolonization plays a determining role in their presence. The SCP is defined as the temperature at which a liquid, such as the insect’s body fluids, transitions into a solid state or crystallizes, releasing the latent heat of crystallization [16,18]. Ice formation within cells causes damage, and if it reaches a certain severity, the insect will not survive upon thawing.

Where HGA occurs, the implementation of resistant cultivars is a valuable management strategy. Despite research efforts directed towards the identification of HGA-resistant wheat [19,20] and HGA-resistant barley genotypes [21], no sorghum cultivars have been identified that provide resistance to HGA. Greenbug-resistant barley cultivars were also identified as being resistant to HGA [22]. The discovery of resistance to HGA in existing aphid-resistant sorghum cultivars offers an opportunity for the rapid deployment of HGA resistance strategies, if needed.

The objectives of this study were to investigate the effects of temperature on the development and reproduction of HGA, to determine the SCP for HGA life stages, and to evaluate the HGA life history on a sorghum-resistant cultivar and different cereal grain hosts compared with the sorghum aphid (SA), *Melanaphis sorghi* (Theobald, 1904).

## 2. Materials and Methods

### 2.1. Aphid Culture

The HGAs utilized in all the experiments were obtained from crested wheat grass *(Agropyron cristatum*) in Taos, New Mexico in 2019 and were maintained on “Yuma” wheat in 4.4 L pots, which were fitted with Lexan™ cylinders (45 cm tall × 16 cm diameter; SABIC Polymershapes, Tulsa, OK, USA) and organdy cloth tops to prevent the escape of the aphids. Colonies of SA were originally collected from a post-harvested grain sorghum field in Matagorda County, TX in 2015 and kept in similar pots to those used for HGA rearing. The SA colonies were maintained on the susceptible sorghum variety TX7000. Infested plants were replaced every week to 2 weeks and watered when needed.

The HGA colonies were maintained in a APHIS-approved quarantine room located on the premises of the USDA-ARS station in Stillwater, OK to prevent any external contamination, as HGA have not been documented in wild populations in Oklahoma [2]. To ensure the colony’s health and survival, the aphids were transferred to new plants every week or as necessary due to deteriorating plant quality. The quarantine room was maintained at a consistent temperature of 21 ± 2 °C and a 14:10 L:D photoperiod with lighting provided by seven TS 32W Ecolux^®^ daylight fluorescent lamps (Fairfield, CT, USA).

### 2.2. Experiment to Determine HGA Development under Different Temperatures

Six identical growth chambers (Percival^®^ Model E30B, Perry, IA, USA) that provided temperature, light, and humidity control were used in this experiment. Each growth chamber was considered as one treatment, and in each, the programmed temperatures were 10, 15, 20, 25, 30, and 35 ± 0.5 °C with a 14:10 L:D cycle. Each growth chamber supplied identical light provided by eight fluorescent grow lights (Philips Inc., Guadalajara, Mexico).

The wheat variety “Jagger” used in this experiment was planted in Cone-tainers™ (model SC10; S7S Greenhouse Supply, Tangent, OR, USA) containing three layers of media: potting soil, fritted clay, and sand (bottom to top, respectively). Each Cone-tainer™ was fitted with an 8 cm diameter Lexan sleeve that was 45 cm in height and ventilated with organdy cloth. After the plant reached approximately 5 cm in height, 10 sexually mature adult aphids were placed on each plant using a horsehair paintbrush.

The aphids were left on the plants for 24 h to produce offspring, and all but one, the 1st instar, were removed from the plant. This 1st instar aphid was left on the plant and checked every 24 h until reproduction to determine the pre-reproduction period (d). After the aphid became sexually mature, each nymph produced was counted and removed. Data were collected every 24 h until the death of all the original females.

Because the experimental unit in this study consisted of climate-controlled chambers, each plant with a single female was considered as a pseudo-replicate, and there were twelve pseudo-replications for each temperature treatment. This approach was adopted in order to use the insects in the same generation, considering a limited number of climate-controlled chambers. Consequently, a conservative statistical approach was adopted when conducting treatment comparisons.

### 2.3. Supercooling Point Experiment

To determine the supercooling point (SCP) of the HGA, individuals were attached to a 30-gauge, copper-constant thermocouple coated with a thin layer of petroleum jelly (Walmart Inc., Bentonville, AR, USA). The thermocouple was attached to a CR12x micrologger (Campbell Scientific, Logan, UT, USA) and placed inside a Pyrex^®^ test tube measuring 2.5 × 15 cm. The test tube was held erect and partly submerged in a dry ice (≈0.5 kg) and a 70% ethyl alcohol (≈0.75 L) bath following the methodology described by Armstrong et al. [16]. The bath temperature was maintained between −30 °C and −35 °C, and the exposure temperature was controlled by lowering the thermocouple down into the test tube. The temperature of the thermocouple was recorded every 0.2 s using a data logger. The lowest body temperature of each individual was identified by plotting its temperature time series, where the release of the latent heat of crystallization was determined, known as the SCP. Individuals of each developmental stage (1st instar, apterous adult, and winged adult) were used in the study, and each individual was considered as one replicate. Twelve individuals were measured for each developmental stage.

### 2.4. Comparison of the Life History of HGA and SA on Different Hosts

Three sorghum genotypes, including the known SA-resistant TX2783 and two known susceptible TX 7000 and KS 585 [23,24] hybrids, a wheat (Custer) and a millet (Millex 32), were used for assessing the life parameters of SA and HGA. Each plant species was first infested with 3–5 adult reproductive females, which were left for 24 h. After this 24 h period, all the aphid adults and all but one nymph were removed from the plants. The aphids and plants were then placed in two identical growth chambers (Percival^®^ Model E30B, Perry, IA, USA), set at a constant temperature of 23 °C, a 14:10 day–night cycle, and a relative humidity (RH) of approximately 70%. The aphids were observed every 24 h, and data were collected from the day they began to reproduce until their death. Each plant with a single female was considered a replication, with eight replications for each aphid species and on each host plant.

### 2.5. Statistical Analysis

The experiments were conducted using a completely randomized design. The demographic parameters evaluated included fertility, longevity (days), pre-reproductive period (d), number of progeny produced in an equal period d(Md), and the intrinsic rate of increase (rm) calculated using the equation: rm = 0.74 (logeMd)/d following the method described by Wyatt and White (1977) [25,26]. In the SCP experiment, we determined the supercooling points of each individual and then calculated the average for each life stage. We compared the average supercooling points between different life stages.

In addition, the relationship between temperature (T) and developmental rate (r = 1/d) for the pre-reproductive time was modeled using linear regression, using the formula Dr = a + bT, where Dr is the development rate (days−1) and T is the temperature (°C) [27] within a temperature range in which the relationship was linear (15 to 30 °C). The lower temperature threshold (T0) and the thermal constant (K, degree-days) were determined using the parameters: T0 = −a/b and K = 1/b [28].

The statistical analyses were conducted in the R computing environment, utilizing the “AgroR” package [29] for the analyses and the “ggplot2” package [30] for creating graphs. Before proceeding with the main analyses, we performed exploratory data analysis to assess the assumptions of the normality of residuals [31] and homogeneity of variances [32]. The means were analyzed using a one-way ANOVA with a significance level of α = 0.05.

To address the pseudo-replication issue in the temperature trials, we employed Tukey’s post hoc test. This well-established method is known for its conservative approach to separating treatment means [33]. In the plant host experiment, we used the *t* tests to compare aphid species on the same host, setting the significance level at α = 0.05.

## 3. Results

### 3.1. HGA Development at Different Temperatures

Temperatures of 10 °C and 35 °C did not allow for the development of HGA until reproduction. Development and reproduction occurred within the temperatures of 15 °C, 20 °C, 25 °C, and 30 °C (Table 1). The aphids reared under temperatures of 25 °C and 30 °C had similar pre-reproductive periods (d; F = 175.64, DF_numdf;demdf_ = 3; 41; *p* < 0.001), requiring approximately 10.5 days to produce their first offspring (Table 1). At 25 °C, the HGA had the highest intrinsic rate of increase (rm) of 0.245 ± 0.011, while those reared at 15 °C showed the lowest rm of 0.085 ± 0.005 (Table 1). The linear equation describing the pre-reproductive period (d) in relation to temperature (°C) was as follows: Dr = 0.0041T−0.0172; (R^2^_adj_ = 0.80; *p* > 0.001). The lower temperature threshold (T0) was 4.2 °C, and the thermal constant (K, degree-days) was 243.9.

The females reared at a temperature of 30 °C exhibited a significant reduction in the number of progenies produced (Md), producing approximately one-third of the offspring compared to those reared at 20 °C and 25 °C. These females also had the lowest fertility, producing a markedly smaller number of offspring throughout their lifespans: approximately one-sixth that of the females maintained at 20 °C and 25 °C (Table 1). The females reared at 30 °C also exhibited the shortest lifespans among the viable temperature conditions, with a mean lifespan of 18.50 ± 1.70 days (Table 1).

### 3.2. Supercooling Point

The supercooling point (SCP) for the HGA was similar among the tested life stages. The SCP values of nymphs and apterous adults were −16.55 ± 2.04 °C and −16.25 ± 0.45 °C, respectively, while the winged adults had an SCP of −15.30 ± 0.46 °C (Figure 1). The differences were not statistically significant (F = 0.278, DF_numdf;demdf_ = 2; 57, *p* = 0.758).

### 3.3. Life History of HGA and SA on Different Host Plants

The HGA successfully completed its life cycle on all the tested host plants, with no significant differences in the parameters evaluated (Table 2). The mean number of offspring per female varied from 22 aphids on the Millex32 millet to 51 aphids per female on the Custer wheat. The mean longevity ranged from 32 days on the sorghum TX2783 to 45 days on the sorghum KS585, and the pre-reproductive period (d) was 11 days on the Custer wheat and 13 days on the sorghum KS585. The intrinsic rate of increase (rm) ranged from 0.15 on TX2783 to 0.21 on the wheat Custer.

In contrast, the Custer wheat and Millex32 millet were unsuitable for the SA, with only one individual female reaching the adult stage on each host plant. Therefore, the biological parameters for the SA were compared among the sorghum genotypes (TX7000, TX2783, and KS585) and with the HGA (Figure 2). The SA produced more offspring than the HGA on TX2783; however, the longevity was higher for the HGA than for the SA when KS585 was used as a host (Figure 2A,B). On all the host plants, the SA had a lower pre-reproductive period (d; about half) than the HGA, and they had approximately double the intrinsic rate of increase (rm; Figure 2C,D). Among the sorghums tested, the genotype TX2783, the SA-resistant cultivar, also impacted SA development differently, requiring a longer pre-reproductive period and lowering the rm to 0.12 compared to the other sorghum genotypes.

## 4. Discussion

Although previous studies have reported the effects of temperature [3,34] and the host plant [3] on the development and reproduction of HGA, the present work is the first to investigate the SCP of different stages of HGA development and to evaluate HGA resistance in sorghum SA-resistant cultivars. Our findings contribute to a deeper understanding of the temperature-dependent life history traits of this aphid in the U.S. and improve our understanding of its potential distribution range in temperate areas.

We found that temperatures of 10 °C and 35 °C did not allow the HGA to survive or reproduce. Although HGA is found in a wide range of regions and edaphoclimatic conditions and can utilize a variety of host plants [1,2,3,7,8], during the winter period, HGA populations become scarce [1], possibly because of limited survival during long exposures to low temperatures [1,35]. 

Moreover, the linear equation that described the relationship between the pre-reproductive period and temperature (Dr = 0.0041T−0.0172) estimated a lower temperature threshold (T0) of 4.195 °C; however, the HGA under a constant temperature of 10 °C could not complete their development and did not reproduce.

In South America, HGA is found only in regions with an annual mean temperature > 18 °C, and they are absent in subtropical zones, suggesting that temperature is one important factor for HGA distribution [1,34]. This likely also influences the seasonality of winged forms of HGA, as there is a strong correlation between their peak of flight activity and the mean air temperature [3]. A constant temperature of 30 °C was also highly detrimental to HGA development, survival, and reproduction, suggesting that migration to cooler areas may be a response to increasing temperatures. Previous aphid studies have found that at this temperature, the fitness of species such as the English grain aphid *Macrosiphum avenae* [36] and the corn aphids *Sitobion avenae* and *Metopolophium dirhodum* [26] is considerably affected, with some experiencing 100% mortality.

Lampert et al. (2023) [3] reported similar pre-reproductive periods for HGA when they were reared on wheat (10.6 d), barley (11.7 d), oat (11.1 d), and annual ryegrass (11.1 d) at 25 °C. In contrast, Ricci and Kahan (2005) [37] found that the reproductive period was shorter (9.4 d) when using barley as a host at 20 °C. Female HGA reared on wheat and barley showed a similar longevity compared to our study (46 d and 41 d, respectively) at the same temperature of 25 °C [3]. However, the value of rm found in the present study (0.245) was considerably lower than found in Lampert et al. (2023) [3] (0.79) when using wheat as a host at 25 °C. These differences should be further explored, considering wheat varieties and local populations of HGA to determine reasons for these differences.

Prolonged exposure to temperatures below or above the adaptive range adversely impacts aphid metabolism and energy reserves and subsequently leads to reduced growth and reproductive rates. Additionally, temperatures outside the optimal range can lead to higher mortality rates, ultimately contributing to a decrease in aphid populations even when suitable hosts are present [35,38,39].

Our findings indicate that temperatures between 20 °C and 25 °C are most suitable for HGA development and reproduction. Within this range, HGA females had a higher fertility, longevity, number of progeny (Md), and intrinsic rate of natural increase (rm), as well as a lower pre-reproductive period (d). These demographic parameters suggest that within this temperature range, HGA has a faster life cycle and can reproduce rapidly, potentially leading to large infestations in a short period of time and potentially resulting in damage to crops. These results can be used to develop a predictive model for HGA distribution. On a farm level, these models can also help to predict potential populations and identify the appropriate time to adopt management strategies aimed at minimizing crop loss, should such treatments become necessary.

Determination of the SCP also aids in modeling the distribution, population fluctuation, and minimum survivable temperatures [40,41]. At the SCP, the hemolymph in an insect’s body undergoes a phase change from liquid to solid, forming ice crystals within cells and tissues [16,18,42]. Although some groups of insects are able survive even at temperatures below the SCP [35,38], the formation of ice in tissues is usually lethal, as it causes cellular damage and disrupts physiological processes. It can also impact pest management programs. For example, the adoption of the bacteria *Pseudomonas syringae* as a management tool can lower aphids’ SCP, thereby reducing the temperature at which they freeze and consequently making the aphid more vulnerable to freeze-induced mortality [16].

The SCP is determined by an organism’s inherent traits, such as its body composition, which is closely related to its physiological state (e.g., feeding status, diapause, life stage, metamorphosis) [35]. Our results indicate that first instars, apterous adults, and winged adults of HGA have equivalent SCPs, despite their physiological differences.

However, under field conditions, the HGAs may employ various overwintering tactics to improve their survival, even when temperatures fall below their SCP. One strategy involves seeking refuge under a protective layer of snow, which insulates the aphids from severe cold temperatures. Furthermore, a mutualistic relationship between HGAs and ants could contribute to the aphids’ survival during winter. In some areas, ants gather aphid nymphs and adults and carry them to their subterranean nests, shielding them from harsh winter conditions [43].This symbiotic relationship aids in the survival of both species throughout the winter months and positively impacts their respective populations, although this mutualism has not been reported for HGAs.

Many aphids that are found in temperate zones can overwinter as eggs. However, HGA presents an anholocyclic parthenogenetic form that persists year-round in the tropics and subtropics [44] in its natural range in the Palearctic. HGAs also have a holocyclic lifecycle with overwintering eggs [45,46].

A previous modeling study demonstrated that favorable climatic conditions for HGAs can be found on every continent [8]. However, the predicted HGA distribution model indicates a limited area of the US, covering only the eastern part of the continent around the Great Lakes and along the coasts of Massachusetts and New York, especially on Long Island [8]. Field monitoring conducted from 2015 to 2017 revealed HGA persistence in the central United States, and additional work has not revealed HGA at the sites predicted to be most favorable [2].

In addition to the thermal requirements, understanding the performance of HGAs on different host plants is important to define their potential distribution, especially in large-scale agricultural crops. Our results showed that HGAs can complete their life cycle on all the tested host plants, including the SA-resistant sorghum cultivar (TX2783), showing an equivalent performance. The SA resistance traits from TX2783, which confer antibiosis and tolerance to sorghum for the SA [24], did not show resistance for the HGA. Thus, if HGAs increase in number, others tactics will be required for their management [47]. Our data also suggest that compared with SAs, HGAs have a lower potential to develop on the tested sorghums genotypes. It is also important to consider the limitations of this study, including the focus of the controlled laboratory conditions and the use of a single SA and HGA biotype.

We determined the effects of temperature on life history using constant temperatures maintained in the growth chambers for each temperature rather than in replicated chambers. Previous studies have demonstrated the potential limitation of this approach and additional assessment and refinement of temperature requirements, including the effects of fluctuating temperatures on HGA development, is warranted [48].

In addition, tests of HGA on different grass hosts under different temperatures is warranted [49,50]. The differences between HGA individuals reared in the laboratory and those found in the field [42], as well the potential differences between HGA populations [14], should also be analyzed.

In conclusion, our study sheds light on the crucial role of temperature and the host plant in the development, survival, and reproduction of the HGA. We found that temperatures between 20 °C and 25 °C were optimal for HGA development and reproduction, and the SCPs are similar for different stages of HGAs. The ability of HGAs to develop on all the tested grain crops and the absence of HGA resistance in the SA-resistant cultivar TX2783 highlights the importance of exploring different management strategies. Overall, by expanding our knowledge of temperature-dependent development, reproduction, and mortality of HGAs, along with the exploration of resistant cultivars, we can better predict and manage future HGA populations, ultimately contributing to the sustainable management of this cereal pest.

## Figures and Tables

**Figure 1 insects-14-00862-f001:**
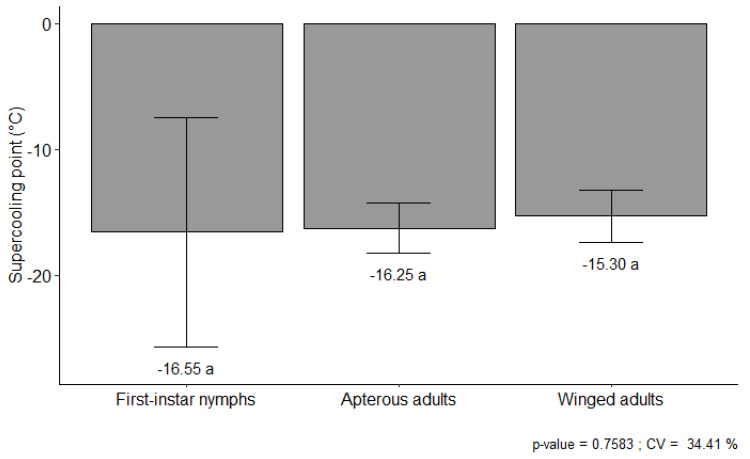
Supercooling point ± SE (°C) of different development stages of the hedgehog grain aphid, *Sipha maydis*, reared under laboratory conditions (21 ± 2 °C and 14:10 L:D). a. The means did not differ significantly based on an F test (F = 0.27; DF_numdf;demdf_ = 2; 57; *p* > 0.05).

**Figure 2 insects-14-00862-f002:**
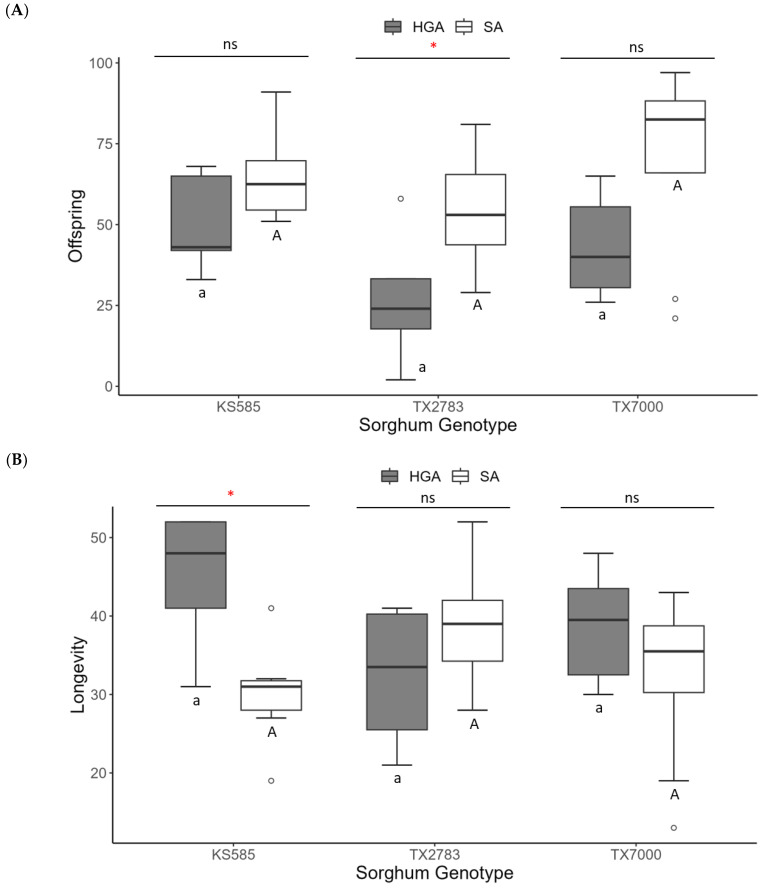
Demographic parameters of the hedgehog grain aphid (HGA) and the sorghum aphid (SA) raised with different sorghum genotypes. (**A**) Number of offspring; (**B**) longevity; (**C**) pre-reproductive period (d); (**D**) intrinsic rate of increase (rm). Box plots followed by the same lower-case letter for HGA and upper-case letter for SA are not significantly different according to a Tukey test (α = 0.05); ns = no significant difference between HGA and SA within the same host according to a *t* test (α = 0.05); * = significant difference (*p* < 0.05) according to a *t* test. The boxplot summarizes: minimum, first quartile (Q1), median (Q2), third quartile (Q3), and maximum. The white dots represent outlier values.

**Table 1 insects-14-00862-t001:** Reproductive parameters of the hedgehog grain aphid, *Sipha maydis* reared at different temperatures in growth chambers with a photoperiod of 14:10 (L:D) and 70 RH ^1^.

Temperature (°C)	Offspring	Longevity (Days)	Pre-Reproductive Period (d) ^2^	Number of Progenies Produce in d (Md)	Intrinsic Rate of Natural Increase (rm)
10	-	-	-	-	-
15	31.46 ± 3.60 b	63.91 ± 4.71 a	28.18 ± 1.15 a	26.09 ± 2.42 a	0.085 ± 0.005 c
20	65.67 ± 4.30 a	57.58 ± 3.14 a	14.08 ± 0.43 b	32.67 ± 2.63 a	0.183 ± 0.009 b
25	63.58 ± 7.39 a	39.83 ± 3.76 b	10.50 ± 0.23 c	33.00 ± 2.83 a	0.245 ± 0.011 a
30	11.00 ± 1.89 c	18.50 ± 1.70 c	10.50 ± 0.22 c	10.20 ± 1.57 b	0.153 ± 0.018 b
35	-	-	-	-	-
F	27.40	30.59	175.64	17.45	35.72
DF_numdf;demdf_	3; 41	3; 41	3; 41	3; 41	3; 41
*p*	<0.001	<0.001	<0.001	<0.001	<0.001

^1^ Means ± SE within columns followed by the same letter are not significantly different (Tukey test: *p* > 0.05). ^2^ The linear equation describing the pre-reproductive period (d) in relation to temperature (°C) is as follows: Dr = 0.0041T−0.0172; (R^2^_adj_ = 0.80; *p* > 0.001). The lower temperature threshold (T0) was determined to be 4.195 °C, and the thermal constant (K, degree-days) was found to be 243.902.

**Table 2 insects-14-00862-t002:** Life table of the hedgehog grain aphid (HGA) *Sipha maydis* on different host plants: Sorghum TX7000, TX2783, and KS585, wheat “Custer”, and millet “Millex32”, in growth chambers at a constant temperature of 23 °C, a photoperiod of 14:10 (L:D), and 70% RH ^1^.

Host Plant	Offspring	Longevity (d)	Pre-Reproductive Period (d)	Intrinsic Rate of Natural Increase (rm)
TX7000	37.00 ± 8.32	38.67 ± 2.96	12.17 ± 0.40	0.19 ± 0.01
TX2783	27.00 ± 11.57	32.25 ± 4.92	12.75 ± 1.11	0.15 ± 0.04
KS585	50.20 ± 6.90	44.80 ± 3.99	13.40 ± 0.68	0.18 ± 0.00
Custer	50.71 ± 9.01	40.14 ± 3.81	11.00 ± 0.58	0.21 ± 0.02
Millex32	21.71 ± 7.44	37.20 ± 6.01	11.60 ± 0.51	0.17 ± 0.00
F	1.62	0.94	2.28	1.91
DF_numdf;demdf_	4; 22	4; 22	4; 22	4; 22
*p*	0.20	0.46	0.09	0.15

^1^ Means ± SE within columns did not differ significantly based on an F test (*p* > 0.05).

## Data Availability

The data presented in this study are available on request from the corresponding author.

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
