# Peer review of "Effects of Temperature and Host Plant on Hedgehog Grain Aphid, Sipha maydis Demographics"

_insects, 2023, doi:10.3390/insects14110862_

Round 1
Reviewer 1 Report
Comments and Suggestions for Authors
Interesting experimental work on the new for the USA pest Sipha maydis . A very well undertaken experiment to investigate SCPs!
Comments:
- the aphid in its natural range (Palearctic) is a holocyclic species. The surviving form is the overwintering egg. Please refer to this in the discussion.
- line 264 - the cited work [5] shows such a model, it can be discussed in relation to the results obtained and real locations of this species in the USA
- can S. maydis and the second examined species, Sorghum aphid, feed on the same host plant, i.e. can they jointly weaken crop plants or are they vicarians?
- line 16 Together these results help us predict and manage HGA populations in cereal crops - please discuss
- line 281 - are there known hibernation tactics of S. maydis as an anholocyclic species?
- to what extent S. maydis feeding on wild grasses, which constitute its reservoir, is recognized in the USA?
Author Response
- The aphid in its natural range (Palearctic) is a holocyclic species. The surviving form is the overwintering egg. Please refer to this in the discussion.
Added: L2L322-325: Many aphids that occur in temperate zones can overwinter as eggs. Although HGA presents an anholocyclic parthenogenetic form that persists year-round in the tropics and subtropics [44], in its natural range in the Palearctic HGA. has a holocyclic lifecycle with overwintering eggs [45,46].
2) line 264 - the cited work [5] shows such a model, it can be discussed in relation to the results obtained and real locations of this species in the USA
added: L326-332: A previous modeling study demonstrated that favorable climatic conditions for HGA can be found on every continent [8]. However, the predicted HGA distribution model indicates a limited area of the US, covering only the eastern part of the continent around the Great Lakes and along the coasts of Massachusetts and New York, especially on Long Island [8]. Field monitoring conducted from 2015 to 2017, reveals HGA persistence in the central United States and additional work has not revealed HGA at the sites predicted be most-favorable [2].
3) can S. maydis and the second examined species, Sorghum aphid, feed on the same host plant, i.e. can they jointly weaken crop plants or are they vicarians?
The reviewer asks a good question. Few studies have examined multi-aphid interactions on a host plant and to the best of our knowledge, there is no scientific record of the interaction between S. maydis and Melanaphis sacchari in terms of host response
4) line 16 Together these results help us predict and manage HGA populations in cereal crops - please discuss
L17 edited: Together these results help us predict HGA populations by identifying host plants in cereal crops and documenting temperature effects on populations.
5) line 281 - are there known hibernation tactics of S. maydis as an anholocyclic species?
As far we know, there is no description of hibernation/diapause in S. maydis
6) to what extent S. maydis feeding on wild grasses, which constitute its reservoir, is recognized in the USA?
L43-49 edited: HGA was first detected in the United States in 2007, where it was found feeding on giant wild rice, Leymus condensatus [7]. The most recent field assessment, conducted from 2015 to 2017, revealed that HGA has since expanded and occurs in New Mexico, Colorado, Utah, and Wyoming where it uses a range of grass species [2].
Reviewer 2 Report
Comments and Suggestions for Authors
The authors have done a good study. The experiments were well designed and conducted. The statistical analysis is correct. The text is well and clearly written. The results of the study are not of high fundamental value but can be used for the elaboration of the methods for prediction and control of an important cereal pest, the hedgehog grain aphid. Thus, the manuscript can be published although some improvements are still required before publication (see me comments below). Some of these improvements could be rather time-consuming and therefore my decision is ‘major revision’.
Major comments
Based on your data (Table 1), the rate of maturation (reciprocal of the duration of the pre-reproductive period) linearly depends on temperature in the range from 15 to 25 C and then stay stable up to 30 C. This pattern of thermal impact is typical of insects and other ectotherms; a number of linear and non-linear models were used for its description by different researchers (e.g. Jafari et al., Modeling Thermal Developmental Trajectories and Thermal Requirements of the Ladybird Stethorus gilvifrons. Insects 2023, 14, 11). These models, in particular, allow to predict the pre-reproductive period at various (not tested) temperatures. Thus, I would strongly recommend treating your data with some of these models and selection of the best fitting one.
Similarly, it could be interesting to plot other studied parameters shown in Table 1 against temperature, to estimate the general pattern of dependence, and to determine the appropriate regression models.
Minor comments
Line 94: Really? The whole laboratory (not only the quarantine room) was maintained at these constant temperature and photoperiod?
Lines 100-101: Please indicate the accuracy of temperature control in the growth chambers.
Figure 2. Please, explain in the legends what is shown in the box plots: median, quartiles, range and outliers? Possibly, some of the potential readers are not familiar with this standard.
Author Response
Reviewer #2
- Based on your data (Table 1), the rate of maturation (reciprocal of the duration of the pre-reproductive period) linearly depends on temperature in the range from 15 to 25 C and then stay stable up to 30 C. This pattern of thermal impact is typical of insects and other ectotherms; a number of linear and non-linear models were used for its description by different researchers (e.g. Jafari et al., Modeling Thermal Developmental Trajectories and Thermal Requirements of the Ladybird Stethorus gilvifrons. Insects 2023, 14, 11). These models, in particular, allow to predict the pre-reproductive period at various (not tested) temperatures. Thus, I would strongly recommend treating your data with some of these models and selection of the best fitting one.
We appreciate the reviewer’s suggestion. We adopted the linear model to get the equation and to estimate the T0 (lower temperature threshold) and K(degree-days) (Jafari et al, 2023)
L167-172 added: In addition, the relationship between temperature (T) and developmental rate (r = 1/d) for the pre-reproductive time was modeled using linear regression, using the formula Dr = a + bT, where Dr is the development rate (days−1), and T is the temperature (°C) [27], within the temperature range in which the relationship was linear (15 to 30°C). The lower temperature threshold (T0) and the thermal constant (K, degree-days) were determined through the parameters: T0 = −a/b and K = 1/b [28].
2) Similarly, it could be interesting to plot other studied parameters shown in Table 1 against temperature, to estimate the general pattern of dependence, and to determine the appropriate regression models.
Done!: The linear equation describing the pre-reproductive period (d) in relation to temperature(°C) is as follows: Dr = 0.0041T-0.0172; (R2adj= 0.80; p>0.001). The lower temperature threshold (T0) is determined to be 4.2°C, and the thermal constant (K, degree-days) is found to be 243.9.
3) Line 94: Really? The whole laboratory (not only the quarantine room) was maintained at these constant temperature and photoperiod?
The word “laboratory” was changed to “quarantine room” in L98, in order to avoid misunderstanding.
4) Lines 100-101: Please indicate the accuracy of temperature control in the growth chambers.
Done! ±0.5°C
5) Figure 2. Please, explain in the legends what is shown in the box plots: median, quartiles, range and outliers? Possibly, some of the potential readers are not familiar with this standard.
A brief explanation was added: The boxplot summarizes: minimum, first quartile (Q1), median (Q2), third quartile (Q3), and maximum. The white dots represent outlier values.
Reviewer 3 Report
Comments and Suggestions for Authors
This study by Mason Taylor and colleagues explores key factors influencing hedgehog grain aphid (HGA) development, survival, and reproduction, a cereal pest. HGA first appeared in the US in 2007, prompting a need for population insights for distribution predictions and effective management. The study focuses on temperature and host plants, revealing that 20-25°C is ideal for HGA development and reproduction. Temperatures below 10°C and above 35°C hinder HGA survival. Additionally, researchers examine supercooling points for different HGA life stages and compare HGA demographics with sorghum aphids on various hosts, finding HGA can complete its cycle on resistant wheat, millet, and sorghum.
The experimental design appears to be appropriate, but the description lacks clarity and I have several questions regarding experimental design, sample sizes, and analysis. These are listed below:
The study's analysis exhibits notable limitations, especially in its employment of ANOVA within the framework of intricate experimental designs. The suitability of this statistical method for the study's design warrants careful consideration. It is imperative to offer transparent elucidations regarding the rationale for selecting these statistical methods and how potential challenges stemming from the lack of statistical independence among multiple measurements of offspring developmental times from individual female subjects were mitigated. Relying solely on Tukey tests may prove insufficient to address this independence issue, suggesting the potential need for more robust and intricate statistical models.
Furthermore, it's important to recognize that fecundity is likely impacted by the longevity of adult females. Consequently, it is strongly advised to incorporate longevity as a covariate in the analysis to investigate this relationship effectively. A mere declaration that all data were subjected to ANOVA (or GLMs or GLMMs) is inadequate. There is a pressing need to clearly delineate both the dependent and independent variables while expounding on any data transformations or the absence thereof. The current manuscript lacks clarity concerning the specific parameters under investigation and whether the data adhered to a normal distribution. Given the current structure and content, questions may arise regarding the suitability of ANOVA as an approach. Therefore, a revision of both the Results and Discussion sections is recommended to ensure alignment with these considerations.
Moreover, the Results section necessitates enhancement in terms of clarity and brevity, focusing exclusively on the findings derived from this particular study. The article would significantly benefit from a more comprehensive exploration of hypotheses and expectations, forging connections with prior studies, particularly those encompassing field research or investigations into the repercussions of fluctuating temperatures on the species under study and related species. These treatments are more pertinent to the environmental conditions experienced in natural settings. The authors should explicitly delineate the motivations behind their research, its significance, and its alignment with previous studies. The inclusion of this information holds the potential to substantially elevate the overall quality of the paper.
I hope you will consider revising the content in line with the ideas I've expressed here and then resubmit.
Author Response
Reviewer #3
1) The study's analysis exhibits notable limitations, especially in its employment of ANOVA within the framework of intricate experimental designs. The suitability of this statistical method for the study's design warrants careful consideration. It is imperative to offer transparent elucidations regarding the rationale for selecting these statistical methods and how potential challenges stemming from the lack of statistical independence among multiple measurements of offspring developmental times from individual female subjects were mitigated. Relying solely on Tukey tests may prove insufficient to address this independence issue, suggesting the potential need for more robust and intricate statistical models.
We have addressed the lack of independence of the replications adopting the Tukey post-hoc test since it is considered a conservative test (Lee and Lee, 2018), in order to keep a balance between statistic power and type I error rate. We also have addressed the pseudo-replication of our treatments, since we adopted a growth chamber for each temperature.
2) Furthermore, it's important to recognize that fecundity is likely impacted by the longevity of adult females. Consequently, it is strongly advised to incorporate longevity as a covariate in the analysis to investigate this relationship effectively. A mere declaration that all data were subjected to ANOVA (or GLMs or GLMMs) is inadequate. There is a pressing need to clearly delineate both the dependent and independent variables while expounding on any data transformations or the absence thereof. The current manuscript lacks clarity concerning the specific parameters under investigation and whether the data adhered to a normal distribution. Given the current structure and content, questions may arise regarding the suitability of ANOVA as an approach. Therefore, a revision of both the Results and Discussion sections is recommended to ensure alignment with these considerations.
Thank you for your valuable insights. Fecundity is closely connected to the longevity of females, which is why we have adopted the life table parameters developed by Wyatt and White (1977) and used since. This method has been widely adopted in entomology studies, particularly in research on aphids (Cao et al. 2015; Dampc et al. 2021; Mornhinweg et al. 2020; Paudyal et al. 2019b, 2019a). Considering an organism that reproduces parthenogenetically, this method accounts for the pre-reproductive period (i.e., the number of days before the first reproduction, d), the number of progeny produced within d (Md), and subsequently calculates the intrinsic rate of natural increase (rm). The analysis of these parameters provides valuable insights into potential population growth within a defined time window, and these parameters are amenable to comparison through ANOVA. While offspring numbers are undeniably linked to longevity, the lifetable parameters offer a robust tool for estimating population growth over a specified duration (Wyatt & White 1977).
3) Moreover, the Results section necessitates enhancement in terms of clarity and brevity, focusing exclusively on the findings derived from this particular study. The article would significantly benefit from a more comprehensive exploration of hypotheses and expectations, forging connections with prior studies, particularly those encompassing field research or investigations into the repercussions of fluctuating temperatures on the species under study and related species. These treatments are more pertinent to the environmental conditions experienced in natural settings. The authors should explicitly delineate the motivations behind their research, its significance, and its alignment with previous studies. The inclusion of this information holds the potential to substantially elevate the overall quality of the paper.
Thank you for your valuable comments and suggestions. As the comments were similar to those of reviewer 1, and stressed connection to existing literature and brevity, we added material as detailed above:
L 180-183 added: The linear equation describing the pre-reproductive period (d) in relation to temperature(°C) was: Dr = 0.0041T-0.0172; (R2adj= 0.80; p>0.001). The lower temperature threshold (T0) was 4.2°C, and the thermal constant (K, degree-days) was 243.9.
L223-226: Moreover, the linear equation that described the relationship between pre-reproductive period and temperature (Dr = 0.0041T-0.0172) estimated the lower temperature threshold (T0) of 4.195°C, while HGA under constant temperatures of 10°C couldn't complete its development and did not reproduce.
L 268-272 added: Additionally, another overwintering strategy adopted by some aphids is to pass through it in the egg stage. Although HGA presents an anholocyclic parthenogenetic form that persists year-round in the tropics and subtropics [34], in its natural range in the Palearctic, Sipha sp. primarily has a holocyclic form and one survival strategy adopted by some species is overwintering eggs [35,36].
L273-278: A previous modeling study demonstrated that favorable climatic conditions for HGA can be found on every continent, performing better in a wider climatic range [8]. In North America, the predicted HGA distribution model indicates a limited area of the US, covering only the eastern part of the continent around the Great Lakes and along the coasts of Massachusetts and New York, especially on Long Island [8]. However, according to the latest field monitoring conducted from 2015 to 2017, HGA still hasn’t reached these sites [2].
Round 2
Reviewer 2 Report
Comments and Suggestions for Authors
Thank you for considering all my comments.
Reviewer 3 Report
Comments and Suggestions for Authors
The authors have effectively addressed my comments, and no further action is required.